

# Role of ELK1 in regulating colorectal cancer progression: miR-31-5p/CDIP1 axis in CRC pathogenesis

Guoqiang Yan and Lei Wang

Department of Colorectal & Anal Surgery, The First Hospital of Jilin University, Changchun, Jilin, China

## ABSTRACT

**Background and Objective**. Colorectal cancer (CRC) is a malignant tumor that affects the digestive system. With the increased of modernization of society, the incidence of colorectal cancer has increased throughout the world. As a transcription factor, ELK1 has been widely studied in colorectal cancer. However, there are still many unknown factors regarding its specific mechanism of action. This study explored the role of ELK1 and its downstream pathway in CRC pathogenesis.

**Methods**. Based on clinical samples, this study examined miR-31-5p expression in CRC cells and its impact on malignant behaviors (migration, invasion, apoptosis) and autophagy. The promoter sequence of miR-31-5p was obtained from the UCSC database, and ELK1 was identified as its transcription factor. In ELK1-knockdown CRC cells, miR-31-5p was overexpressed, and its response in malignant behaviors and autophagy was analyzed. The target gene CDIP1 was predicted and verified using a dual-luciferase assay. The influence of CDIP1 on malignant behavior in CRC cells was assessed, and CDIP1 siRNA was used as a rescue treatment for miR-31-5p inhibition. The role of ELK1/miR-31-5p in tumor growth was validated *in vivo*.

**Results**. miR-31-5p expression was upregulated in the colorectal cancer tissues and cells. The knockdown of miR-31-5p markedly inhibited cancer cells' malignant behaviors and mediated autophagy. ELK1 was confirmed to bind with the miR-31-5p promoter and enhance miR-31-5p transcription. miR-31-5p was found to bind with the CDIP1 3'UTR and inhibit CDIP1 expression. CDIP1 siRNA partially rescued the effects of miR-31-5p knockdown on cell metastatic ability, autophagy, and apoptosis. Based on the *in vivo* experiments, results showed that the ELK1/miR-31-5p axis positively regulated tumor growth in nude mice.

**Conclusion**. Our findings indicate that ELK1 regulates the progression of colorectal cancer *via* an miR-31-5p/CDIP1 axis, and the ELK1/miR-31-5p/CDIP1 axis could be a therapeutic target for colorectal cancer.

## INTRODUCTION

Colorectal cancer (CRC) is a malignant tumor that originates from the colorectal mucosa and is the most frequently seen malignancy that affects the digestive system (*Xue et al., 2018*;

Corresponding author
Lei Wang, leiwang1967@126.com

*Siegel, Miller & Jemal, 2018*). As more people have adopted western diets and lifestyles, the incidence of CRC has increased, and the numbers of young people with CRC continues to rise (*Brody, 2015*; *The Lancet Oncology, 2017*). The 5-year survival rate of CRC patients in low-income countries is <50%, and slightly >10% among patients with stage IV disease (*Brenner, Kloor & Pox, 2014*). Thus, developing an understanding of the genetic changes and molecular mechanisms involved in CRC progression may provide new strategies for the early diagnosis of CRC and targeted therapy.

ELK1, as a part of Ets family of transcription factors, contains a conserved DNA-binding domain of Ets that is accountable for transcription at Ets-sequence containing promoters. ELK1's activation domain plays a crucial role in enabling the protein to execute its physiological functions (*Morris et al., 2013*), and plays a carcinogenic role in many cancers, including colorectal cancer (*Xu et al., 2018*; *Ma et al., 2021b*; *Wang et al., 2020*). For example, ELK1 interacts with the androgen receptor (AR) to enhance the expression of a significant portion of AR-regulated genes predominantly associated with cell growth processes in prostate cancer (*Patki et al., 2013*; *Rosati et al., 2018*). As for colorectal cancer, ELK1 expression was found to be upregulated in tumor-associated macrophages and associated with poor survival in CRC patients (*Wang et al., 2020*). The downstream ELK1 becomes activated upon triggering c-KIT signaling. Thus, activated P-ELK1 increases the transcription of carcinoembryonic antigen, leading to enhanced cell adhesion, migration, and invasion in CRC (*Ma et al., 2021b*). Phosphorylation at the T417 site of ELK1 is observable in the nuclei of epithelial cells in multiple normal and cancerous tissues. There is a positive correlation between the number of pT417-positive cells and the differentiation stage of colonic adenocarcinoma (*Morris et al., 2013*). ELK1 can alter the transcription of LBX2 AS1, and thus foster the advancement of colorectal cancer (*Ma et al., 2021a*). Therefore, the specific molecular and cellular mechanism of ELK1 in colorectal cancer needs further exploration.

MiR-31-5p is considered to be an oncomiR in colorectal cancer. Abundant expression of this molecule is observed in colorectal cancer and it has been shown to target NUMB, thereby exerting a significant influence on a variety of cellular events, such as proliferation, cell cycle progression, migration, invasion, and apoptosis (*Peng et al., 2019*). An upregulation of miR-31-5p results in colorectal cancer cell resistance to oxaliplatin (*Hsu et al., 2019*). In patients with colon adenocarcinoma, by targeting TNS1, miR-31-5p is considered a feasible prognostic factor that has been observed to affect immune infiltration (*Mi et al., 2020*). In CRC patients treated with anti-EGFR therapeutics, a high level of miR-31-5p expression is associated with shorter progression-free survival, and this implies that miR-31-5p could potentially serve as a valuable supplementary prognostic biomarker in the context of anti-EGFR therapy (*Igarashi et al., 2015*). While miR-31-5p's function has been previously documented, the interplay between ELK1 and miR-31-5p in the context of colorectal cancer remains unexamined.

In the present study, we were aiming to (1) explore the role of ELK1 and its downstream pathway in the pathogenesis of colorectal cancer (CRC), and evaluated the impact of the ELK1/miR-31-5p pathway on tumor formation and investigated the potential of the ELK1/miR-31-5p/CDIP1 axis as a therapeutic target for colorectal cancer. This study will

discover that ELK1 regulates the progression of colorectal cancer *via* the miR-31-5p/CDIP1 axis and will reveal that the ELK1/miR-31-5p/CDIP1 axis could be a potential therapeutic target for colorectal cancer, thereby providing a new research direction and basis for the treatment of colorectal cancer.

## MATERIALS AND METHODS

### Tumor tissue obtaining

Paired normal and tumor tissues were collected from colorectal cancer patients diagnosed at The First Hospital of Jilin University. Patients who had received any previous chemoradiotherapy or targeted therapy were excluded from this study. The Ethics Committee of The First Hospital of Jilin University granted approval for tissue sampling (2022-KS-080). All participating patients offered their written informed consent to partake in the study.

### Cell lineage and cell culture

Colorectal cancer cell lines, RKO (ATCC® CRL-2577), LoVo (CCL-229™), HT29 (HTB-38™), Coco-2 (HTB-37™), and NCI-H498 (CCL-254™), as well as FHC cell line (CRL-1831™) were obtained from ATCC (Manassas, VA, USA). EMEM and Hybri-Care Medium, purchased from ATCC, was used for culture of colorectal cancer cells, supplemented with 10% FBS and 1% Penicillin-Streptomycin. The FHC cells were cultured in MEGM Kit medium (Catalog No. CC-3150; Lonza, Hayward, CA, USA) supplemented with 10% FBS. All cells were cultured at 37 °C in a 5% $CO_2$ atmosphere.

### Real-time quantitative polymerase chain reaction (qRT-PCR)

Total RNA was extracted from tissues or cells in compliance with the manufacturer's guidelines. A reverse transcriptase kit (Invitrogen) with oligo (dT, 20 primer) was utilized for cDNA synthesis. Ultimately, SYBR green PCR Master Mix (Qiagen, Hilden, Germany) was employed in conjunction with quantitative PCR instrument (Stratagene, Mx3000P) provided by Agilent Technologies (Agilent Technologies Inc., Carpinteria, CA, USA) for conducting PCR. The reaction mix underwent an initial denaturation at 95 °C for 2 min, followed by 40 cycles of amplification, consisting of denaturation at 94 °C for 20 s, annealing at 58 °C for 20 s, and extension at 72 °C for 20 s. GAPDH was used to normalize mRNA expression levels, and U6 was used as the internal reference to miRNA expression. Table 1 displays the primers employed for qRT-PCR. The $2^{-\Delta\Delta Ct}$ method was applied to determine relative mRNA and miRNA expression levels.

### TdT-mediated dUTP nick end labeling (TUNEL) assay

To detect apoptotic cells in tumor tissues, sections of consecutive tissue were subjected to TUNEL staining that was performed using an *in-situ* cell death detection kit (Beyotime, Jiangsu, China). The tumor tissue sections embedded in paraffin underwent triple washes in 0.01 M PBS and were subsequently permeabilized with proteinase K for 10 min. Following another round of three washes, the sections were placed in a buffer consisting of TdT and FITC-labeled dUTP mixture, and incubated in the dark at 37 ° C for 1 h. Finally,

Table 1  Primer sequence used in PCR.

| ID | Sequence (5′–3′) | Product length (bp) |
|---|---|---|
| GAPDH F | TGTTCGTCATGGGTGTGAAC | 154 |
| GAPDH R | ATGGCATGGACTGTGGTCAT | |
| ELK1 F | TACCTCCACCATGCCAAATG | 167 |
| ELK1 R | TGAAGGTGGAATAGAGGCCC | |
| CDIP1 F | TACATGCCTCCGGGGTTTCTA | 267 |
| CDIP1 R | ACCCAGCACGAAATTCATCA | |
| U6 F | CTCGCTTCGGCAGCACA | 96 |
| U6 R | AACGCTTCACGAATTTGCGT | |
| All R | CTCAACTGGTGTCGTGGA | |
| miR-31-5p | AGGCAAGATGCTGGCATAGCT | |
| miR-31-5p RT | CTCAACTGGTGTCGTGGAGTCGGCAATTCAGTTGAGAGCTATG | |
| miR-31-5p F | ACACTCCAGCTGGGAGGCAAGATGCTGGCAT | |

DAPI was employed as the counterstaining agent, and the sections were observed under a fluorescence microscope.

## Western blot assay

The total proteins were obtained from samples and its concentration was detected using the bicinchoninic acid protein assay (BCA; Pierce, Rockford, IL, USA). Subsequently, a 20 μg portion of protein from each extract underwent separation *via* Tris-glycine SDS-PAGE (4% to 20%). Afterwards, protein bands were transferred to PVDF membranes (Sigma-Aldrich, St. Louis, MO, USA), followed by a blocking with non-fat milk. Following this, the membranes experienced an 8∼10 h incubation with primary antibodies, including anti-LC3B (ab51520, Abcam, Cambridge, MA, USA), anti-Beclin 1 (ab207612, Abcam, Cambridge, UK), anti-P62 (ab109012, Abcam, Cambridge, UK), and anti-GAPDH (ab8245, Abcam, Cambridge, UK). They were subsequently incubated with the secondary antibodies (BA1055 for rabbit-sourced primary antibody, and BA1051 for mouse-sourced primary antibody, Boster, Pleasanton, CA, USA). Bands were visualized using an Enhanced Chemiluminescence Kit (ECL; Pierce, Appleton, WI, USA). Expression of GAPDH was used as internal reference to targeting proteins.

## Cell transfection

To accomplish the knockdown of ELK1, plasmids designed with short hairpin RNA targeting ELK1 (siRNA ELK1) were utilized (GenePharma, Sunnyvale, CA, USA). Lipofectamine®3000 reagent, provided by Invitrogen was employed to complete the cell transfection. Post-transfection, cells were collected after 48 h for further analysis.

## Wound healing assay for cell migration

Target cells were added to each well of a 24-well plate ($5\times 10^5$ cells per well) and allowed to grow for 24 h. Next, a pipette tip was used to make one scratch across the cell monolayer, with the pipette being kept as perpendicular to the horizontal line as possible. The cells were

then washed three times with PBS, fresh complete medium was added, and the cells were incubated for an additional 24 h at 37 ° C in a 5% $CO_2$ atmosphere. Following incubation, the width of the scratch was measured, and pictures were taken.

## Cell invasion detection (Transwell)

Subsequent to suspending cells in medium deprived of serum, they were added into the upper compartment within a Transwell plate (Jet Bio, Guangzhou, China) that was previously plated with Matrigel Matrix (BD Biosciences; San Jose, CA, USA). The lower chamber of the culture was supplemented with medium that contained 10% FBS, and incubated for 24 h. After incubation, the cells on the membrane's upper surface were removed, whereas the cells located on the under-surface membrane were fixed using 4% PFA for 10 min before being incubated with 0.4% crystal violet solution. Lastly, invasive cells were imaged utilizing a digital microscope produced by Olympus (Tokyo, Japan).

## Promoter sequence acquisition and transcription factor prediction

Using the UCSC database, we obtained the promoter sequence of miR-31-5p ( −5000 bp). Subsequently, we identified potential transcription factors that may bind to the miR-31-5p promoter sequence by utilizing the JASPAR database at http://jaspar.genereg.net/. The binding site of ELK1 on miR-31-5p sequence was shown in (Supplementary File S1).

## Identifying miR-31-5p target genes

To identify the target genes of miR-31-5p, Targetscan online tool (https://www.targetscan.org/vert_80/) was used with a criterion of Total context++ score less than −0.2, resulting in the identification of 358 genes. These genes were then categorized into three groups: oncogenes, anti-oncogenes, and unknown factors. Focusing on anti-oncogenes, the Kaplan–Meier Plotter online tool (http://kmplot.com/analysis/index.php) was employed to conduct survival analysis, in order to select factors significantly associated with colorectal cancer prognosis. Further clinical sample validation was performed on the selected factors, and the gene with the lowest expression in cancer tissues was chosen as the target gene.

## Dual-luciferase reporter assay

The binding site of miR-31-5p on the CDIP1 3′-UTR was predicted using Targetscan. The CDIP1 3′-UTR sequence containing miR-31-5p binding sites cloned using PCR with primers (forward: TATCCTGCCACTGTCCTTCC, reverse: CCCGACTCTGGTTTG-GTTTA). To create a wild-type reporter vector (wild type, WT), the amplified product was inserted into the psi-check2 vector (Promega, Madison, WI, USA). Furthermore, luciferase reporters with the predicted binding sites of miR-31-5p in the CDIP1 3′-UTR sequence were created to generate a mutant-type reporter vector (mutant type, MUT). Lipofectamine 3000 was employed to co-transfect the miR-31-5p mimics/miR-31-5p inhibitor in the reporter vectors into 293T cells, and the Dual-Luciferase Reporter System (Promega, Madison, WI, USA) was used to measure luciferase activity.

## ChIP assay

The EZ-Magna-ChIP HiSens Kit (17-10461; Burlington, MA, USA) protocol was followed to conduct ChIP assays. In brief, RKO cell extracts were sonicated three times for ten

cycles of 30 s ON/30 s OFF on ice using the Bioruptor® PLUS sonication device set at the HIGH-power position (H). Immunoprecipitation (IP) of cross-linked protein/DNA complexes was accomplished using magnetic protein A/G beads. ELK1 antibodies (1:25) or control IgG (1:25) were used to immunoprecipitate the cleaved DNA. The fragments of the miR-31-5p promoter contained in the precipitated DNA were detected using qRT-PCR, and the fold enrichment was calculated in comparison to the input DNA.

## Establishment of a xenograft nude mouse model

Six-week-old female BALB/c nude mice were purchased from the Animal Center of Jilin University, and subsequently housed under specific pathogen-free conditions (22 ° C, ~50% humidity). Twelve mice were randomly assigned to 4 separate groups: Blank group, NC group, ELK1 siRNA group, and ELK1 siRNA+miR-31-5p. RKO cells were treated with ELK1 siRNA and miR-31-5p mimics. After being treated for 48 h, RKO cells were harvested and subcutaneously injected into the right anterior armpit of each nude mouse, with a total infusion volume of 50 μL of PBS and a concentration of $1\times 10^6$ RKO cells. Tumor growth was monitored over the next 28 days, and the weight of the tumors was recorded following the sacrifice of the mice. The animal studies were performed in compliance with the guidelines established by The First Hospital of Jilin University Animal Care and Use Committee (Approval No. 20200712).

## Hematoxylin & eosin (H&E) staining

H&E staining was performed according to the description by *Li et al. (2021)*. Briefly, the sections were immersed in hematoxylin staining solution for a 15-min duration, followed by exposure to eosin solution. Once gradient alcohol dehydration and xylene transparency treatments were performed, the tissue sections' pathological features were examined using a microscope (Olympus, Tokyo, Japan).

Initially, tissue specimens were embedded in paraffin, from which 4-μm thick slices were obtained and subsequently dewaxed using dimethylbenzene.

## Statistical analysis

IBM SPSS Statistics for Windows, Version 20 software (IBM Corp, Armonk, NY, USA) was utilized for the statistical analysis of all data sets, and the results are presented as the mean value ± standard deviation (SD). The Shapiro–Wilk test was used to determine the normal distribution of the data. To compare the data between two groups, Student's *t*-test was employed, while one-way ANOVA followed by Dunnett's T3 test was utilized for analyzing differences between more than two groups. For non-parametric data, the Whitney U test was utilized. A *P*-value below 0.05 indicated a statistically significant difference.

## RESULTS

### Expression characteristic of miR-31-5p in CRC tissues and cells

To verify the expression of miR-31-5p in colorectal cancer, we collected samples of CRC and adjacent normal intestinal tissue and determined their levels of miR-31-5p expression *via* qRT-PCR (Fig. 1A). The results showed that miR-31-5p was expressed at significantly
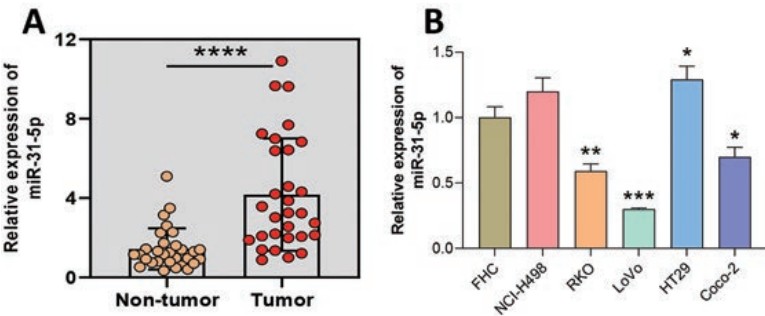

**Figure 1** **Aberrantly high expression of miR-31-5p in human colorectal cancer tissues and cells.** (A) The levels of miR-31-5p expression in collected human colorectal cancer tissues and adjacent normal intestinal mucosa tissues were detected by qRT-PCR. (B) The levels of miR-31- 5p expression in the human normal colon epithelial cell line, FHC, and colorectal cancer cell lines, NCI-H498, RKO, LoVo, HT29, and Coco-2, were detected by qRT-PCR. *$P < 0.05$, **$P < 0.01$, ***$P < 0.001$.

higher levels in the tumor tissues than in the normal tissues. Next, miR-31-5p expression in FHC, and the CRC cell lines (NCI-H498, RKO, LoVo, HT29, and Coco-2) was examined by qRT-PCR (Fig. 1B). The results showed that miR-31-5p was expressed at much higher levels in two of the colorectal cancer cell lines (RKO and LoVo) than in the normal colon epithelial cells.

## Inhibition of miR-31-5p affected CRC line metastasis, autophagy, and apoptosis

Considering miR-31-5p's overexpression in CRC clinical tissues and cells, we investigated its implications in colorectal cancer cells. We effectively inhibited miR-31-5p in RKO and LoVo cells by treatment with miR-31-5p inhibitor, a procedure which was further validated by qRT-PCR analysis (Fig. 2A). Results showed that miR-31-5p altered cell migration (Fig. 2B), cell invasion (Fig. 2C), cell autophagy (Fig. 2D), and cell apoptosis-related proteins of CRC cells (Fig. 2E) were detected, respectively. Our results showed that inhibition of miR-31-5p decreased cell migration and invasion, and increased the levels BAX protein (Fig. 2F) and also the levels of caspase 3, caspase 9, caspase 11 activity (Figs. 2G, 2H, 2I) in both of the colorectal cancer cell lines. As for cell autophagy, inhibition of miR-31-5p dramatically increased cell autophagy.

## ELK1 bound to the miR-31-5p promoter and mediated *miR-31-5p* gene transcription

To explore the effects of miR-31-5p in colorectal cancer, we first examined whether the transcription factor ELK1 binds with miR-31-5p. Lysates from CRC cells were mixed with biotinylated probes against the miR-31-5p promoter, and immunoblotting was used to detect the levels of ELK1 protein. EMSA assays showed that the biotin-labeled miR-31-5p promoter could bind with ELK1 protein, while the mutated miR-31-5p promoter failed to pull down ELK1 protein (Fig. 3A). Lysates of colorectal cancer cells were subjected to immunoprecipitation (IP) with antibodies against human IgG and anti-ELK1, and then analyzed by qRT-PCR by using primers specific for the miR-31-5p promoter. ChIP assays

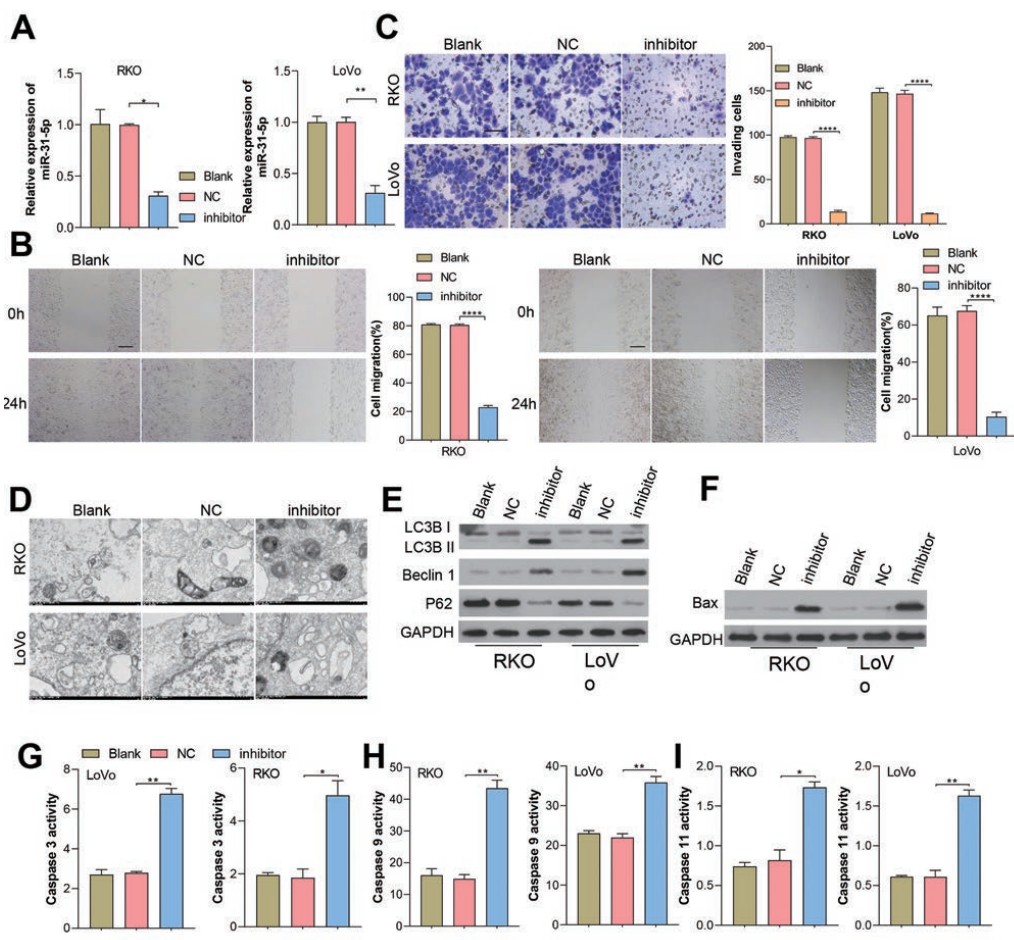

**Figure 2** **Inhibition of miR-31-5p affected colorectal cancer cell metastasis, autophagy, and apoptosis.** (A) miR-31-5p inhibition was achieved in RKO and LoVo cells by transfection with an miR-31-5p inhibitor. The migration abilities of RKO and LoVo cells with miR-31-5p inhibition were detected by wound healing assays (B, scale: 100 $\mu$m), and their invasion capabilities were assessed by the Transwell assay (C, 50 $\mu$m). (D, E) Electron microscopy (D) and immunoblotting (E) were used to detect autophagy. (F, G, H, I) The levels of BAX protein were detected by immunoblotting (F), and caspase 3 (G), caspase 9 (H), and caspase 11(I) activity in RKO and LoVo cells was detected by ELISA. $^{*}P < 0.05$, $^{**}P < 0.01$, $^{***}P < 0.001$.

also showed binding between ELK1 and the miR-31-5p promoter (Fig. 3B). Additionally, as shown in Fig. 3C, ELK1 overexpression significantly increased the transcriptional activity of miR-31-5p, while ELK1 overexpression did not alter luciferase activity in miR-31-5p promoter mutated cells. These results suggested that ELK1 binds to the miR-31-5p promoter and mediates its transcription.

## Effect of ELK1/miR-31-5p axis on CRC cell phenotypes

As aforementioned, ELK1 regulates *miR-31-5p* gene transcription by recognizing and binding with its promoter. The dynamic effects of ELK1 and miR-31-5p were explored *in vitro*. We treated RKO and LoVo cancer cells with ELK1 siRNA and miR-31-5p mimics. As

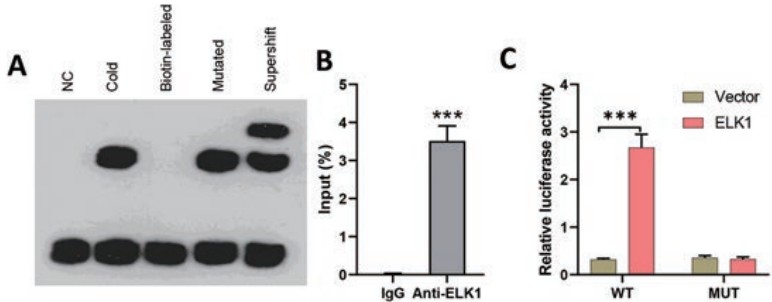

**Figure 3** **ELK1 bound to the miR-31-5p promoter and mediated miR-31-5p transcription.** (A) Lysates from colorectal cancer cells were mixed with biotinylated probes against the miR-31-5p promoter or an oligo. Immunoblotting was used to detect the levels of ELK1 protein. (B) Lysates of colorectal cancer cells were subjected to immunoprecipitation (IP) with antibodies against human IgG and anti-ELK1, followed by qRT-PCR analysis using primers specific for the miR-31-5p promoter. (C) After mutating the miR-31-5p promoter, transcriptional activity of the miR-31-5p promoter was detected in the vector-treated and ELK1-overexpressing cells. $*P < 0.05$, $**P < 0.01$, $***P < 0.001$.

shown in Fig. 4A, miR-31-5p expression was significantly lower in the ELK1-interferenced cells than in the siRNA control cells, indicating that the miR-31-5p mimic significantly increased miR-31-5p expression in the ELK1-interferenced cells. ELK1 expression was significantly lower in ELK1-interferenced cells than in siRNA control cells, and miR-31-5p mimics failed to alter ELK1 expression in the ELK1-interferenced group (Figs. 4A, 4B). Next, cell migration (Fig. 4C), cell invasion (Fig. 4D), and cell autophagy activity (Figs. 4E and 4F), as well as the levels of cell apoptosis-related proteins (Fig. 4G), and caspase 3 (Fig. 4H), caspase 9 (Fig. 4I) and caspase 11 (Fig. 4J) were measured, respectively. Knockdown of ELK1 significantly decreased cell migration and cell invasion, while miR-31-5p overexpression dramatically attenuated the effects of ELK1 interference. In addition, inhibition of ELK1 dramatically promoted cell apoptosis activity and autophagy, while miR-31-5p overexpression partially rescued the effects of ELK1 knockdown on cell apoptosis and autophagy.

## MiR-31-5p bound to the CDIP1 3'-UTR

To explore the mechanism underlying the function of miR-31-5p in colorectal cancer progression, we used the online website TargetScan (http://www.targetscan.org/vert_71/) to predict the binding site between miR-31-5p and the CDIP1 3'-UTR. To confirm the binding site, we constructed two different types of CDIP1 3'-UTR luciferase reporter vectors; wild-type, and mutant-type, respectively (Fig. 5A). Our results showed that overexpression of miR-31-5p markedly downregulated, whereas miR-31-5p inhibition upregulated the luciferase activity of the wild-type-CDIP1 3'-UTR vector, while MUT (mutant type) of binding site on 3' UTR of CDIP1 failed to alter luciferase activity (Fig. 5B).

## Overexpression of CDIP1 affected colorectal cancer cell metastasis, autophagy, and apoptosis

CDIP1 expression was overexpressed in RKO and LoVo cells, and subsequent immunoblotting confirmed the CDIP1 overexpression efficiency (Fig. 6A). After

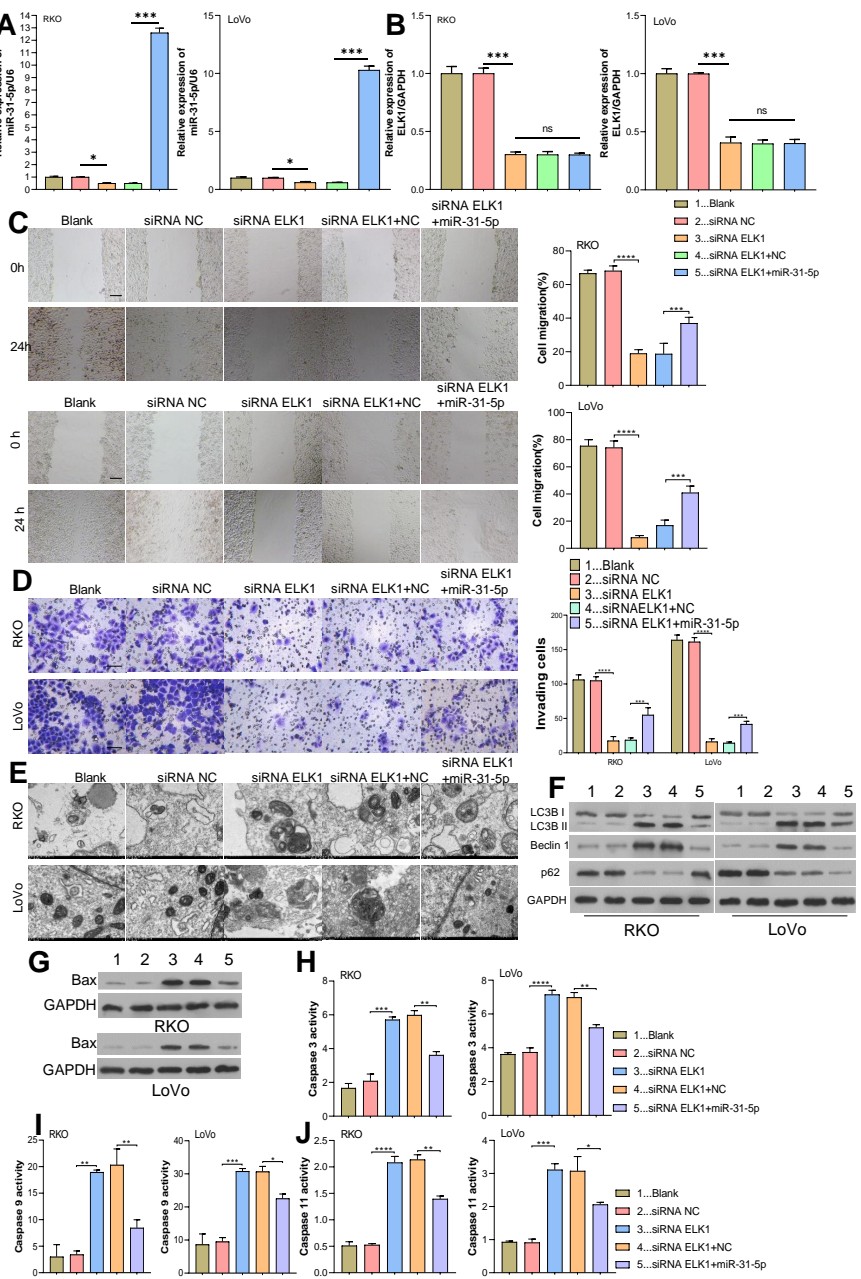

**Figure 4** **Effects of ELK1 and miR-31-5p on colorectal cancer cell phenotypes.** RKO and LoVo cells were transfected with ELK1 siRNA or miR-31-5p mimics. (A) miR-31-5p expression was detected by qRT-PCR. (B) ELK1 expression was detected by qRT-PCR. (C) Cell migration was assessed by a wound healing assay. Scale: 100 μm; (D) cell invasion was detected by the Transwell assay, Scale: 50 μm. (E, F) Electron microscopy (E) and immunoblotting (F) were used to detect autophagy. (G, H, I, J) BAX protein levels were detected by immunoblotting (G), and the levels of caspase 3(H), caspase 9 (I), and caspase 11 (J) activity in RKO and LoVo cells were detected by ELISA. *$P < 0.05$, **$P < 0.01$, ***$P < 0.001$.

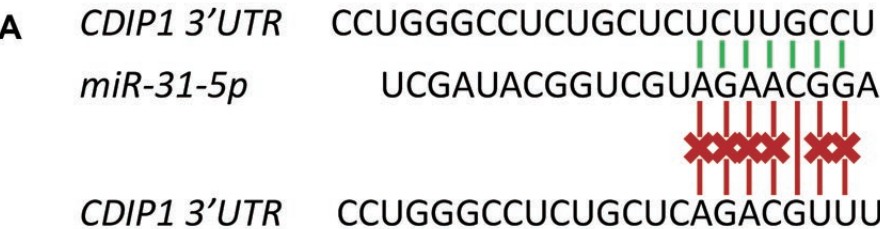

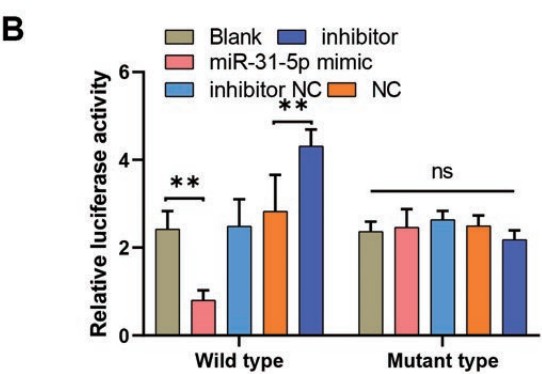

**Figure 5** **MiR-31-5p bound to the CDIP1 3′-UTR.** (A, B) A dual-luciferase reporter assay was performed by constructing wild-type and mutant-type CDIP1 3′-UTR reporter vectors and co-transfecting them into 293T cells along with miR-31-5p mimics or the miR-31-5p inhibitor. Luciferase activity was then determined. $**P < 0.01$.

transfection for 48 h, the levels of caspase 3, caspase 9, and caspase 11 activity (Figs. 6B–6D) in colorectal cancer cells were measured. Those results showed that overexpression of CDIP1 increased the levels of caspase 3, caspase 9, and caspase 11 activity. Next, assays for cell migration (Fig. 6E), cell invasion (Fig. 6F), and cell autophagy (Fig. 6G) were conducted, and the results showed that CDIP1 inhibited the invased and migrated of CRC cells. Cell autophagy was detected by TEM, along with western blotting assay, and results showed that CDIP1 overexpression promoted cell autophagy (Figs. 6G and 6H).

### Dynamic effects of miR-31-5p and CDIP1 siRNA on colorectal cancer cell phenotypes

Since miR-31-5p binds to the CDIP1 3′-UTR, while whether miR-31-5p/CDIP1 alter CRC cell phenotypes remain unknown.

After transfection for 48 h, miR-31-5p expression was measured by qRT-PCR (Fig. 7A). As shown in Figs. 7B and 7C, we found that inhibition of miR-31-5p suppressed the migration and invasion of colorectal cancer cells, and those effects were reversed by CDIP1 siRNA. Furthermore, these results, including TEM and western blotting assay showed that the promotive effect of the miR-31-5p on autophagy could be reversed by CDIP1 siRNA (Figs. 7D and 7E). Additionally, as shown in Figs. 7F and 7G-7I, both Bax expression, and caspase 3, caspase 9, and caspase 11 activity were enhanced in the inhibitor-treated cells. However, the pro-apoptotic role of the miR-31-5p inhibition was abolished partially by treatment with CDIP1 siRNA (Figs. 7F–7G).

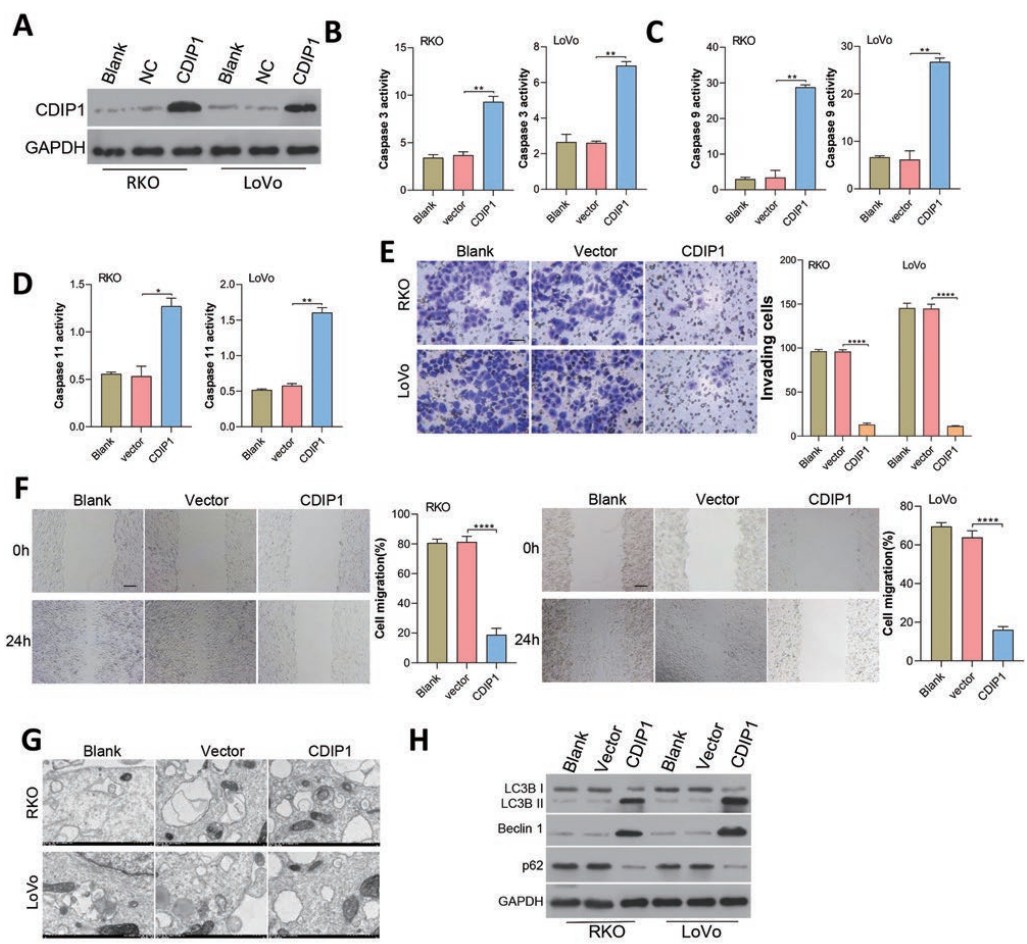

**Figure 6  Overexpression of CDIP1 affected colorectal cancer cell metastasis, autophagy, and apoptosis.** (A) CDIP1 overexpression was achieved in RKO and LoVo by transfection with CDIP1 plasmids. (B, C, D) BAX protein levels detected by immunoblotting, and the levels of caspase 3, caspase 9, and caspase 11 activity in RKO and LoVo cells were detected by ELISA. (E, F) The invasion capabilities of RKO and LoVo cells with CDIP1 overexpression were detected by the Transwell assay (E, Scale: 50 µm) and their migration abilities were detected by the wound healing assay (F, Scale: 100 µm). (G, H) Electron microscopy (G) and immunoblotting (H) were used to detect autophagy. $^*P < 0.05$, $^{**}P < 0.01$, $^{***}P < 0.001$, and $^{****}P < 0.0001$.

## Dynamic effects of ELK1 and miR-31-5p on the tumorigenesis ability of colorectal cancer cells in nude mice

To further confirm the *in vivo* effects of ELK1 and miR-31-5p on colorectal cancer, we established a subcutaneous xenograft tumor mouse model. Colorectal cancer cells were subcutaneously injected into nude mice. As shown in Fig. 8, tumor growth was attenuated by ELK1 siRNA, and the attenuation effect was reversed by the miR-31-5p mimics (Figs. 8A–8C). H&E staining showed that in the ELK1 siRNA-treated tumor tissues, there were larger areas of necrotic tissue that showed positive TUNEL staining, while miR-31-5p treatment reduced the areas of necrotic tissue and shrunk the TUNEL-positive areas

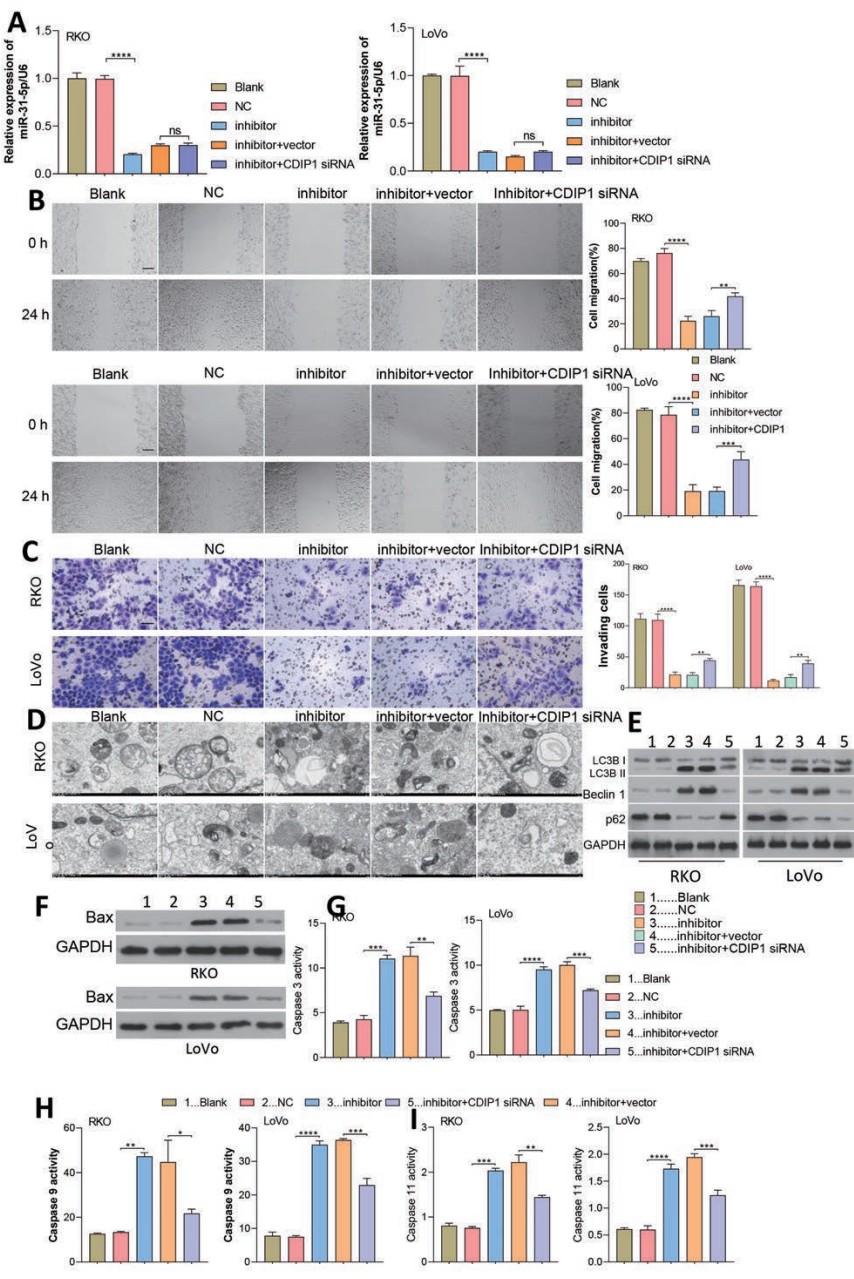

**Figure 7** **Dynamic effects of miR-31-5p and CDIP1 on colorectal cancer cell phenotypes.** (A) After being transfected with miR-31-5p mimics or CDIP1 siRNA, RKO and LoVo cells were analyzed for their miR-31-5p levels by qRT-PCR. The migration ability of transfected RKO and LoVo cells was evaluated by the wound healing assay (B, Scale: 100 $\mu$m), and their invasion capabilities were analyzed by the Transwell assay (C, Scale: 50 $\mu$m). (D, E) Electron microscopy (D) and immunoblotting (E) were used to detect autophagy. (F, G, H, I) Western blotting was used to measure Bax protein expression, and ELISA was used to detect caspase 3 (G), caspase 9 (H), and caspase 11 (I) activity in RKO and LoVo cells. $^*P < 0.05$, $^{**}P < 0.01$, $^{***}P < 0.001$, and $^{****}P < 0.0001$.

(Figs. 8D and 8E). ELK1 expression was obviously suppressed in the tumors of animals in the ELK1 siRNA group, which caused a downregulation of miR-31-5p expression and upregulation of CDIP1. Treatment with miR-31-5p mimics significantly down-regulated CDIP1 expression level in the tumor tissues; however, it had no significant effect on ELK1 expression (Figs. 8F–8H). This result was confirmed by western blotting as shown in Fig. 8I. Furthermore, ELK1 siRNA induced increases in caspase 3, caspase 9, and caspase 11 activity, while miR-31-5p suppressed the activity of those proteins (Figs. 8J–8L). When taken together, our *in vivo* experiments demonstrated how the effects of ELK1 and miR-31-5p interactions affected the tumorigenesis ability of colorectal cancer cells.

## DISCUSSION

Colorectal cancer is the third most common malignancy and one of the leading causes of cancer-related death worldwide (*Brenner, Kloor & Pox, 2014*). The aberrant expression of miRNA suggests its potential use as a diagnostic, prognostic, and predictive biomarker in cancer (*Iorio & Croce, 2012*). This study investigated the role of ELK1 and its downstream pathway in colorectal cancer (CRC) pathogenesis, and found that ELK1 regulates CRC progression *via* an miR-31-5p/CDIP1 axis. The ELK1/miR-31-5p/CDIP1 axis could be a potential therapeutic target for colorectal cancer treatment.

miRNAs serve as critical factors in determining tumorigenicity by distinct mechanisms (*Bartel, 2009*). miR-31-5p has been widely researched in CRC. For example, the levels of serum miR-31-5p are notably elevated in patients with oral cancer. With its ability to enhance oral cancer malignancy in both *in vitro* and *in vivo* models, circulating miR-31-5p holds potential to act as an independent diagnostic biomarker and a therapeutic target for oral cancer treatment (*Lu et al., 2019*). The overexpression of miR-31-5p has been reported to lead to oxaliplatin resistance in CRC cells, indicating that miR-31-5p may play an innovative factor facilitating tumor development (*Hsu et al., 2019*). In this study, we revealed that miR-31-5p was expressed at high levels in colorectal tumor tissues and cancer cells, and that high levels of miR-31-5p abundance in patients with CRC. These results hinted miR-31-5p inhibition markedly decreased the metastatic ability of CRC cells and accelerated their apoptosis and autophagy *in vitro*. These results suggested the significantly detrimental effects of miR-31-5p in colorectal cancer.

ELK1, a transcription factor belonging to the ETS family, interacts with specific DNA sequences and modulates gene expression, activate a series of proto-oncogenes, such as c-Fos (*Hanlon, Bundy & Sealy, 2000*). Within the context of colorectal cancer (CRC), ELK1 plays a role in a multitude of signaling pathways and biological processes, including cell proliferation and apoptosis. Liu et al. find that circ_0022340 targets miR-382-5p to up-regulate the ETS transcription factor ELK1, promoting colorectal cancer progression. This indicates the importance of the ELK1 in colorectal cancer development and suggests a potential therapeutic target (*Liu et al., 2022*). In general, ELK1 serves as a transcription factor and typically functions by regulating the transcription of target genes. Li and colleagues' research demonstrates that ELK1 modulates the m6A reader protein YTHDF1, promoting prostate cancer development and progression through the PLK1/PI3K/AKT

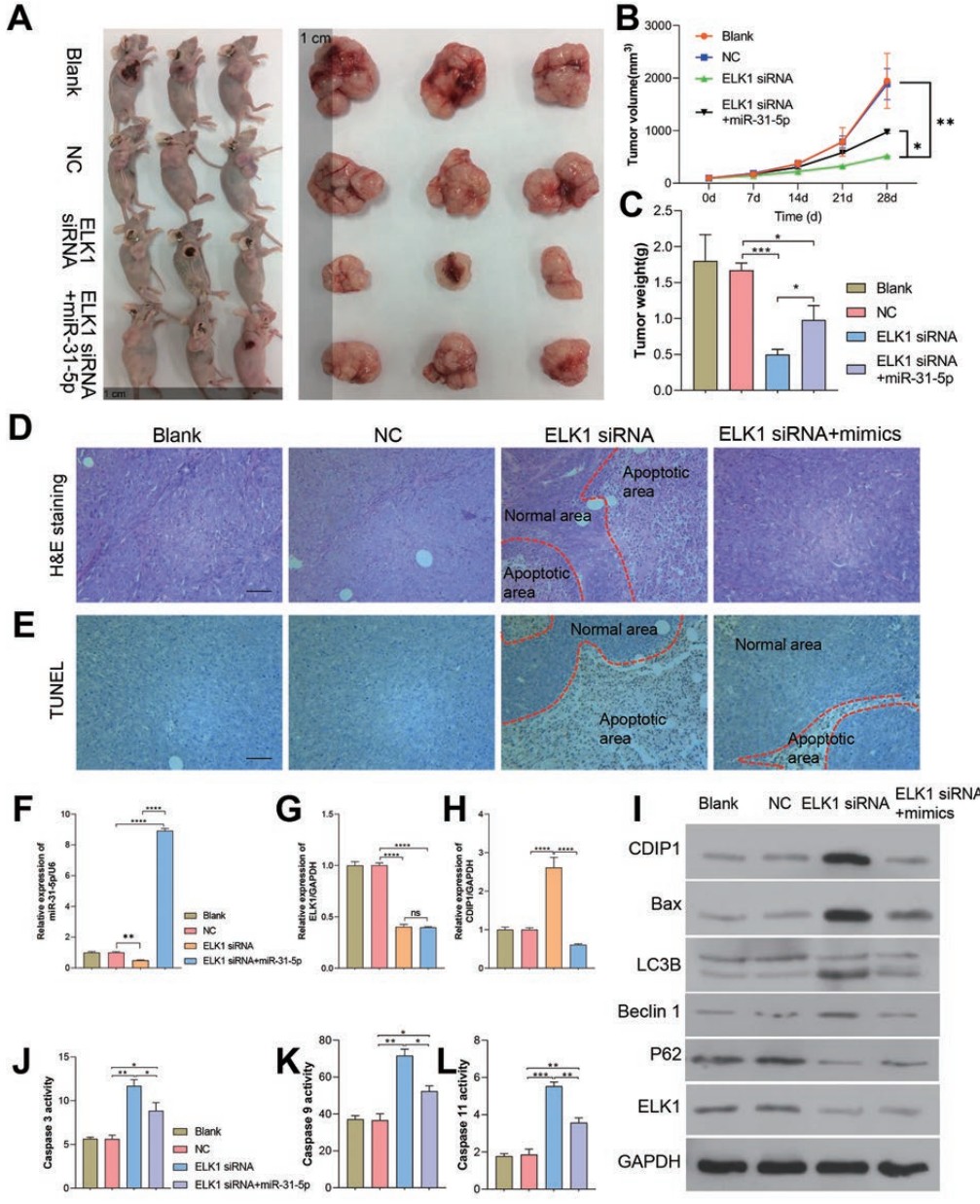

**Figure 8   Dynamic effects of ELK1 and miR-31-5p on the tumorigenesis ability of colorectal cancer cells in nude mice.** A subcutaneously implanted tumor model was established in nude mice by injecting the mice with RKO cells transfected with ELK1 siRNA or miR-31-5p mimics. (A, B, C) The tumor volumes and tumor weights were measured at 28 days post-injection. (D, E) The tumor tissues of each group were removed and fixed. H&E staining (D) and TUNEL staining (E) were used to examine tumor histopathology and detect tumor cell apoptosis, respectively. Scale: 100 μm (F, G, H) Total RNA was extracted from tumor tissues and examined for miR-31-5p (F), ELK1 (G), and CDIP1 (H) expression by qRT-PCR. (I) The levels of CDIP1, BAX, and ELK1 proteins in tumor tissues were detected by immunoblotting. (J, K, L) Caspase 3 (J), caspase 9 (K), and caspase 11 (L) activity in tumor tissues was detected by ELISA. $^*P < 0.05$, $^{**}P < 0.01$, $^{***}P < 0.001$, and $^{****}P < 0.0001$.

axis (*Zhang et al., 2022*). In the present study, we found that ELK1 promotes *miR-31-5p* gene transcription, resulting in CDIP1 expression alteration and autophagy happening.

*CDIP1* (Cell Death Inducing P53 Target (1) is a novel pro-apoptotic gene. Under conditions of genotoxic stress or DNA damage, CDIP1 might be targeted by p53 and regulate TNF$\alpha$-induced apoptosis in a p53-dependent manner, which has implications for TNF $\alpha$-based cancer therapeutics, as well as autoimmune diseases (*Brown et al., 2007*; *Brown-Endres et al., 2012*). In this study, we found that CDIP1, regulated by the ELK1/miR-31-5p axis, was closely related to the occurrence of autophagy in colorectal cancer cells. Numerous studies have confirmed that CDIP1 is closely associated with cell apoptosis. *Inukai et al. (2021)* have found that CDIP1 is a pro-apoptotic protein that interacts with the Ca2+-binding protein ALG-2 and facilitates cell death by improving the bond between CDIP1 and ESCRT-I. The expression levels of ALG-2, ESCRT-I subunits, and VAP affect the sensitivity to anti-cancer drugs that are linked to CDIP1 expression. CDIP1, acting as a signal transducer of ER-stress-mediated apoptosis, regulating mitochondrial apoptosis pathway through BAP31 cleavage and Bcl-2 association, results in a transducing apoptotic signals (*Namba et al., 2013*). At present, there is limited research on the connection between CDIP1 and autophagy. However, according to existing literature studies, there appears to be a link between them. It has been found that CDIP1 can form a complex with BAP31, subsequently inhibiting the expression of BCL-2 (*Namba et al., 2013*). As widely known, BCL-2 can bind to Beclin 1, suppressing the occurrence of autophagy (*Pattingre et al., 2005*). We hypothesize that CDIP1 may promote autophagy and induce autophagic cell death. The results of this study serve to confirm this hypothesis.

Our study provides sufficient evidence to further explore the specific effects of ELK1 on colorectal cancer progression. ELK1 promoted miR-31-5p transcription, allowing increasing of miR-31-5p deteriorate CRC cell phenotypic by targeting CDIP1. Furthermore, the study showed that ELK1 regulates autophagy and colorectal cancer progression *via* an miR-31-5p/CDIP1 axis. Thus, targeting the ELK1/miR-31-5p/CDIP1 axis might be a novel approach for treating colorectal cancer.

## Limitations

However, this study has some limitations that should be considered. First, more clinical samples should be collected for detecting expression of the ELK1/miR-31-5p/CDIP1 axis, and the expression correlations should be analyzed. In addition, the correlations between ELK1 expression and various clinicopathologic features deserve further clinical analysis.

### Funding

The authors received no funding for this work.

### Competing Interests

The authors declare there are no competing interests.
## Author Contributions

- Guoqiang Yan analyzed the data, prepared figures and/or tables, authored or reviewed drafts of the article, and approved the final draft.
- Lei Wang conceived and designed the experiments, performed the experiments, authored or reviewed drafts of the article, and approved the final draft.

## Human Ethics

The following information was supplied relating to ethical approvals (Approval NO.: 2022-KS-080):

The institutional review board and Ethics Committee of The First Hospital of Jilin University

## Animal Ethics

The following information was supplied relating to ethical approvals (Approval NO.: 20200712):

The First Hospital of Jilin University Animal Care and Use Committee

## DNA Deposition

The following information was supplied regarding the deposition of DNA sequences:

The sequences are available at TargetScan (Table S1).

## Data Availability

The data is available at figshare: Wang, Lei (2023). Raw data.zip. figshare. Figure. https://doi.org/10.6084/m9.figshare.22322167.v1.

## Supplemental Information

Supplemental information for this article can be found online at http://dx.doi.org/10.7717/peerj.15602#supplemental-information.

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
