# Peer review of "Role of ELK1 in regulating colorectal cancer progression: miR-31-5p/CDIP1 axis in CRC pathogenesis"

_PeerJ, doi:10.7717/peerj.15602_

## Round 0.1 · original submission · Major Revisions

Based on the reviewer's comments, this study could be accepted for publication in PeerJ with major revisions.

Reviewer 1 ·

Basic reporting

Yan et al's manuscript "ELK1 regulates autophagy and progression of colorectal cancer through miR-31-5p/CDIP1 axis in vitro and vivo" has a relatively complete structure, allowing readers to understand the research content. The citation of references is basically reasonable and meets the needs of the content. The sharing of original data meets the basic requirements of the journal, but there are still obvious issues that need to be revised.

1)This article contains a large number of grammatical issues and must be polished by a fluent English speaker;
2)Please check whether the format of the article meets the requirements of PeerJ Journal. Should be maked the necessary adjustments;
3)The referenced citations do not comply with the requirements of the PeerJ journal, and a considerable amount of information (such as page numbers, issue numbers, etc.) is missing. Please make adjustments and add the missing information accordingly.

Experimental design

Colorectal cancer (CRC) is a leading cause of death worldwide. The aim of this study was to investigate the involvement of ELK1 and its downstream pathway in the pathogenesis of CRC. This study revealed that ELK1 regulates CRC progression through miR-31-5p/CDIP1 axis, suggesting that ELK1/miR-31-5p/CDIP1 axis might represent a potential therapeutic target for the treatment of CRC. Clinically, this study provides new insight into the role of ELK1 and miR-31-5p in CRC pathogenesis and may lead to the development of new targeted therapies for CRC patients. From a scientific perspective, the findings of this study contribute to our understanding of the molecular mechanisms underlying the development of CRC, and highlight the importance of ELK1 and miR-31-5p as potential biomarkers for early diagnosis and prognosis of this disease. Based on these results, this study could be accepted for publication in PeerJ journal with major revisions.

1)In the introduction section, try to minimize general knowledge introductions and pay more attention to the content related to this study.

2)In the materials and methods section, the animal source and ethical approval meet the specifications. The provided reagent materials list the origin, brand, and article number in detail. The method information is relatively complete and meets the requirements.

Validity of the findings

3. There are some details issues in the results section:
1)There are no significant marks on the bar chart. It is unclear if statistical analysis has not been done or if it was omitted. Please check;
2)In which cells were the EMSA experiment, CHIP experiment, and dual-luciferase experiment performed? In RKO or LoVo? Please mention this in the corresponding section of the article.
3)There is no indication on some WB bands indicating in which cell line the experiment was carried out. Please check and supplement;
4)In the H&E staining and TUNEL experiments, please mark the corresponding differences in the figure (Fig 8).

Additional comments

4. In the discussion section, please reduce the repetitive descriptions of the results and increase the discussion of the research content.

5. Please add the limitations of this study.

Reviewer 2 ·

Basic reporting

This original manuscript provides a significant contribution to our understanding of the molecular mechanisms involved in autophagy/CRC progression and sheds light on the potential of targeting ELK1/miR-31-5p/CDIP1 for the treatment of CRC.The overall logic of the article is clear and has certain publication value. At the same time, there are also many questions that require significant revisions.
1.It is noted that the manuscript needs careful editing by someone with expertise in technical English editing paying particular attention to English grammar, spelling, and sentence structure so that the goals and results of the study are clear to the reader.
2.It is necessary to check the completeness of the references and the citation specifications.
3. The Figures and Figure legends need to be modified in detail to achieve the normalization of the article's diagrams and figure legends, such as the font format and size appearing in the Figures should be consistent; all diagrams in single Figure should be merged into one diagram of Fig2, Fig4, Fig 7 and Fig 8; In Fig 1, 3 and 4, the description of *P<0.05, **P<0.01 which was not occurred in these Figures was observed while “***” in Fig 1A was not remarked; In Fig 3, there is no Figure 3D, but a caption for Figure 3D appears.
4. The graph in Fig 2, 4, 6, 7 and 8 needs to be scaled.

Experimental design

The study aims to investigate the role of ELK1 and its downstream pathway in the pathogenesis of colorectal cancer. The study used paired tissues from colorectal cancer patients to measure the expression of miR-31-5p. The study found that miR-31-5p expression was high-regulated in colorectal cancer tissues and cells. Knockdown of miR-31-5p inhibited cell migration and invasion, promoted apoptosis and autophagy. ELK1 was confirmed to bind with miR-31-5p promoter and enhance miR-31-5p transcription. miR-31-5p was found to bind with CDIP1 3’UTR and inhibit CDIP1 expression. CDIP1 overexpression partly rescued the effects of miR-31-5p knockdown in cell metastatic ability, autophagy, and apoptosis. This study provides a significant contribution to our understanding of the molecular mechanisms involved in autophagy/CRC progression and sheds light on the potential of targeting ELK1/miR-31-5p/CDIP1 for the treatment of CRC. However, several issues should be addressed and the comments as follows:
1.Studying the mechanism of ELK1 in CRC can help us better understand the pathogenesis of this cancer, and is expected to provide new therapeutic targets and strategies for the treatment of CRC. However, why ELK1 was selected to study need to be emphasized in the Background and objective of Abstract.
2. The authors need to elaborate systematically rather than simply list for the methods used in the present study in the Methods of Abstract.
3. Currently, there are many miRNAs that have been thoroughly studied in CRC, such as miR-144, miR-185-5p and miR-31-5p. Why the author chose miR-31-5p for research needs further explanation. In addition, it is necessary to explain why ELK1-miR-31-5p participate in CRC regulation is that ELK1 regulates the promoter of miR-31-5p and affects the transcription of miR-31-5p not miR-31-5p silence ELK1 expression by binding the 3 'UTR of ELK1.
4. The author should claim that why they investigate the function of ELK1 on autophagy in the progression of colorectal cancer in the Introduction.
5. The method description in this article is too simple, such as the number of clinical samples, the procedures and systems for amplification of qRT-PCR, the constructs used for Dual-luciferase reporter assay and the second antibody information in the western blot. In order to make it easier for other scholars to repeat data or experiments, the author needs to describe the method in more detail all over the manuscript.

Validity of the findings

1. In the manuscript, the authors indicated that “ELK1 binds to the miR-31-5p promoter and mediates miR-31-5p transcription”. The authors need to clarify the sequence of ELK1 binding to the miR-31-5p promoter and the methods that confirmed the binding sequence between ELK1 and miR-31-5p promoter.
2. In Fig. 4A, please indicated the difference between SiRNA NC and SiRNA ELK1+NC group.
3.How many targets of miR-31-5p was obtained predicted by the online website TargetScan (http://www.targetscan.org/vert_71/)? Why the author selected CDIP1 used for further study in the manuscript should be explained.
4.The expression of LC3BI, LC3BII, Beclin 1 and p62 should be detected in vivo.
5.The effect of ELK1/miR-31-5p/CDIP1 axis on autophagy is worth further discussion, which can provide us with a deeper understanding of the regulation of the process of CRC by regulating A-axis influence autophagy.

---

## Round 0.2 · accepted · Accept

The manuscript has been accepted for publication. The citation style will be addressed during production.

Reviewer 1 ·

Basic reporting

Yan et al. 's revision in accordance with the review proposals generally more in place, worthy of praise. The language of the manuscript has been significantly improved and is basically consistent with publication. However, the form of the reference still needs to be revised to make it conform to the conventions of the journal.

Experimental design

no comment

Validity of the findings

no comment

Additional comments

no comment

Reviewer 2 ·

Basic reporting

The author has made appropriate revisions, except for references.
I think the article can be published after revising the literature format according to the PeerJ reference format.

Experimental design

no comment

Validity of the findings

no comment

Additional comments

no comment